# MULTIAGENT SYSTEM FOR LAYER FREE NETWORK

**Hiroki Kurotaki, Kotaro Nakayama & Yutaka Matsuo**
The University of Tokyo
7 Chome-3-1 Hongo, Bunkyo, Tokyo
`{kurotaki, nakayama, matsuo}@weblab.t.u-tokyo.ac.jp`

## ABSTRACT

We propose a multiagent system that have feedforward networks as its subset while free from layer structure with matrix-vector scheme. Deep networks are often compared to the brain neocortex or visual perception system. One of the largest difference from human brain is the use of matrix-vector multiplication based on layer architecture. It would help understanding the way human brain works if we manage to develop good deep network model without the layer architecture while preserving their performance. The brain neocortex works as an aggregation of the local level interactions between neurons, which is rather similar to multiagent system consists of autonomous partially observing agents than units aligned in column vectors and manipulated by global level algorithm. Therefore we suppose that it is an effective approach for developing more biologically plausible model while preserving compatibility with deep networks to alternate units with multiple agents. Our method also has advantage in scalability and memory efficiency. We reimplemented Stacked Denoising Autoencoder(SDAE) as a concrete instance with our multiagent system and verified its equivalence with the standard SDAE from both theoritical and empirical perspectives. Additionary, we also proposed a variant of our multiagent SDAE named "Sparse Connect SDAE", and showed its computational advantage with the MNIST dataset.

## 1 INTRODUCTION

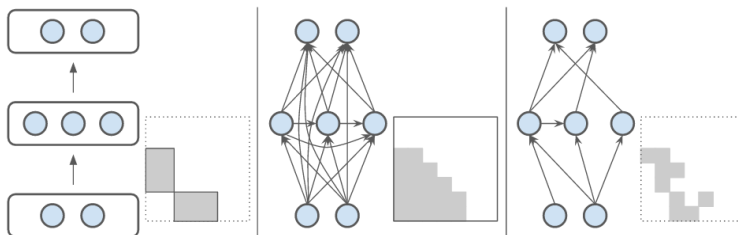

Figure 1: Comparison of network structures with adjacency indications. **Left:** standard feedforward neural network with layer structure. The two solid rectangles represent the weight matrices. The connection is restricted by layer structure. **Center:** layer-free version of standard feedforward network. We need a whole $n \times n$ matrix indicated by the outer solid square to store parameters even when most of them become almost zero as a result of learning. **Right:** an example of network which is a subset of our proposed multiagent system. Matrices (solid rectangle) are no longer used. The network is free from the layer restriction while requiring fewer number of parameters. Communication between the nodes is prohibited unless they are connected with an edge, which leads to scalability, memory efficiency and biological plausibility.

Deep networks are used in many tasks, especially in image recognition, signal processing, NLP and robot manipulation. Almost all deep network models utilize structure with layers. One of the primary reasons to use those layer structure is to utilize parallel computation techniques and hardware level technologies such as GPU or dedicated tensor processors.

Layers can be viewed as a restriction on the connection pattern of the units that they must be aligned in a row to form a vector. Though layers are related to vector-matrix processing technologies and necessary to make use of them, it is not evident that this vector-matrix restriction is the most appropriate model to capture the latent representation of the data.

Vector-matrix approach requires dense matrices to store weight parameters even after sparse representations and weights are learned, which requires inefficient use of memory.

There are several methods to reduce the cost by downsize the model such as distillation(Hinton et al. (2014)), and limiting the model space with binarizing(Courbariaux et al. (2015)) and hashing Han et al. (2015), but they still needs dense matrix filled with many zeros.

In this paper we propose a framework of multiagent calculation network that has feedforward matrix-vector network as a subset but free from the notion of layer to solve these problems.

Our multiagent network consists of many agents that replaces each unit in standard deep networks. The agents act autonomously and calculations are executed as an accumulation of many local level communication among the agents. This local calculation scheme is contrary to the previous feedforward network implementation that all computations in the units in the same layer is done simultaneously (Figure 1).

Specifically, we reimplemented Stacked Denoising Autoencoder(SDAE) (Vincent et al. (2008)) as one of the variations in our framework. SDAE is one of the earliest successful deep network model and free from spatiality assumption in CNNs and RNNs. SDAE is related to the Ladder Network (Rasmus et al. (2015)) in that both employ reconstruction and denoising mechanism. We suppose starting from the model strictly derived from SDAE and gradually extending the model is one good approach to develop useful algorithms to construct deep multiagent networks. We show that the standard vector-matrix SDAE can be interpreted and reconstructed as a specific case of our multiagent network in the sense that both SDAE compute the exact same result in the end but with different implementations and processes.

Once we establish the basic multiagent SDAE, we aim to extend the SDAE and examine its behavior. One of the minimum and simplest modification to the proposed multiagent SDAE is to keep the node's locations and possible edge connections as they are, but randomly truncate the edges. We call this testing model "Sparse connect SDAE" or "SCSDAE" in the latter section. This model is an example of the potential of our model to handle sparse weight parameters more efficient than the standard networks composed of dense weight matrices.

Our contributions are the following.

1. We propose new multiagent-based neural network system framework that free from restrictions of layer scheme with matrix-vector while getting more biologically plausible.
2. We prove that SDAE can be reinterpreted as a special case of our multiagent system.
3. We also propose Sparse Connect SDAE, a primitive extension of our multiagent system.
4. We demonstrate the performance of the proposed models on XOR toy dataset and the permutation-invariant MNIST task.

## 2 RELATED WORKS

Foerster et al. (2016) and Sukhbaatar et al. (2016) model multiagent environment as a deep network, but their unit of agent is a whole network which models an individual actor and not a feature inside the network. The motivation and architecture of our proposal multiagent framework is different from theirs and we suppose our method could be a complementary method between the different granularities of agents.

There are several approaches to design biologically plausible model (Sussillo & Abbott (2009); Risi & Stanley (2014); Mi et al. (2014)). Cao et al. (2014) convert regular CNN into spiking neural networks (SNN). Osogami & Otsuka (2015) build a variation of Boltzman Machine that follows the properties of Spike-timing dependent plasticity (STDP), which makes it more close to biological neural networks. Lee et al. (2014) proposes a modification of autoencoder that uses a novel credit assignment method called "target propagation" in place of back-propagation and achieved state

of the art performance. It can be a solution to remove inherent biologiacal implausibility of back propagation.

Our SCSDAE is a variation of SDAE and can be viewed as a technique for truncating edges and reduce the computational cost. There are several techniques to downsize networks and enables us to load them on mobile devices for realtime processing. One example is distillation (Hinton et al. (2014)). It needs huge networks as a superviser network and it cannot be used as a mean to full scratch to new domain. MADE (Mathieu et al. (2015)) uses binary masking matrix to represent hard zero and thus conditional distribution. Limiting weight connection to binary (Courbariaux et al. (2015)) is another approach and its applicability is actively studied. Han et al. (2015) combines pruning, quantization and Huffman coding together.

## 3 Reimplementation of deep network as a multiagent system

In this section we describe our algorithm. First we show how we can reinterpret some common properties of multiagent systems to form a deep network. Next we define our multiagent system in general form. Then we prove the standard SDAE is indeed a special case of our multiagent system in the sense that both calculate the exact same computational result in the same order. Finally we show an example of our SDAE's variations.

### 3.1 The properties our multiagent system should suffice

We extracted some common aspects of those multiagent system from the view of deep network development as follows:

1. The system consists of several number of autonomous units (as agents) and the environment.
2. All units are autonomous and only act when stimulated by messages from the other units and the environment.
3. Units process calculation only with local information they hold.

### 3.2 Definition of multiagent network

In this section we define the multiagent system that has the properties we stated above.

- A system is consists of several units and one environment.
- Units are consists of nodes and edges.
- Edges connect between two nodes.
- Nodes has some variables and can memorize actual computed values of them.

The variables includes not only the input data and feature variables themselves, but parameters of adjacent edge's weights and intermediate variables that are needed to calculate feature variable's activation values. These additional variables can also be updated.

Nodes can transfer informations with other nodes connected with edges by message passing. These information may involve the value of variable computed by the units, the errors accumulated through epoch in the units, and any other things. Sometimes nodes may receive data input from the environment as a message, and may send value back to environment to let them compute global cost function.

The environment can manipulate unit from outside (via message passing) and change their relationships. For example,

- The environment can generate new unit.
- The environment can connect between nodes.
- The environment can change the state of the unit.
- The environment can send units input data.

These special manipulation may seems to break locality and autonomy, but its global manipulation is limited to changing the overall network structure. In the calculation process, the algorithm still run by agents local algorithm themselves rather than step-by-step instructions from the environment. The environment input some data to input units, and then, all it can do is to wait at the output units to get the result and it isn't aware of the internal behavior of each units and the order of execution.

Algorithm 2 is the general form of our proposed multiagent system. The notation is matched with the multiagent MLP in later sections. Note that the order of calculation is not yet specialized to match to standard MLP, and thus free from the constants related to global structure such as $N_x$ and $N_z$.

## 3.3 THE EQUIVALENCE BETWEEN THE MULTIAGENT AND STANDARD VERSIONS OF SDAE

We show how to reconstruct Stacked Denoising Autoencoder((Vincent et al. (2008))) with the proposed multiagent system.We concurrently prove that our multiagent version of SDAE and the standard SDAE have indeed the same algorithm and do the same calculation. In our case, we suppose that the two criteria below are sufficient for our objective

- the equivalence of the value computed
- the equivalence of the order of computation

given the same input and the same random sampling. (e.g. initial parameters, order of input data, noise variables, etc.)

We begin our proof with a simple model and reuse it for prove of more complex models. Our first target model is the simple multi layer perceptron(MLP) with only one hidden layer and no pretraining. We then go on to the autoencoder and Denoising Autoencoder, which is a kind of extension of MLP. Finally we show the equivalence of the two SDAEs.

For bravity, we limit the choice of activation function for each unit to sigmoid, and we use unit-wise mean square error (MSE) for cost function. We also use simple SGD without minibatch. There are several methods that empirically known to perform well. (e.g. Adam (Kingma & Ba (2014)) for optimization, cross entropy for cost function, ReLU and softmax for activation function.) However we prioritize to establish the basic architecture of deep multiagent network, and choose these simpler methods. We can also extend the algorithm to apply minibatch updating easily.

### 3.3.1 EQUIVALENCE WITH MLP

Now we prove the equivalence of our proposed multiagent version of MLP and standard implementation.

Suppose there is a MLP consists of an input layer, a hidden layer and an output layer. Let $N_x, N_y, N_z$ be the number of input, hidden and output layer's dimension respectively. Simillary, let $x_i (i = 1 \cdots N_x), y_j (j = 1 \cdots N_y), z_k (k = 1 \cdots N_z)$ be the activated output value (hereafter simply "output value" ) at each unit in each layer respectively. The weight value between these variables are $w_{ij}, w_{jk}$, thus the forward propagation from input to hidden is calculated by $y_j \leftarrow \sigma(w_{0j} + \sum_{i=1}^{N_x} w_{ij} x_i)$ and from hidden to output is $z_k \leftarrow \sigma(w'_{0k} + \sum_{j=1}^{N_y} w_{jk} y_j)$. Here $\Theta = \{w_{ij}, w'_{jk} \mid i = 0 \cdots N_x, j = 0 \cdots N_y, k = 1 \cdots N_z\}$ represents the weight parameters and $\sigma$ is the sigmoid function. We also define dataset $\mathbb{D} = \{(x_{data}^{(1)}, t_{data}^{(1)}), \cdots, (x_{data}^{(D)}, t_{data}^{(D)})\}$ as an array of pairs of input data $x_{data}^{(d)} = \{x_{1 data}^{(d)}, \cdots, x_{N_x data}^{(d)}\}$ and label data $t_{data}^{(d)} = \{t_{1 data}^{(d)}, \cdots, t_{N_z data}^{(d)}\}$, where $d = 1 \cdots D$. Hereafter we might drop the data index $(d)$ for readability.

We define our objective function as $L = \sum_{d=1}^{D} L^{(d)}$ , where $L^{(d)} = \frac{1}{N_z} \sum_{k=1}^{N_z} (t_k - z_k)^2$ and $\eta$ be a learning rate. Algorithm 1 is the standard version algorithm to optimize this objective.

Next we construct a multiagent network that is equivalent to Algorithm 1 by specializing the general Algorithm 2. We first generate units $u_{x_i}, u_{y_j}, u_{z_k} (i = 1 \cdots N_x, j = 1 \cdots N_y, k = 1 \cdots N_z)$ from the environment. We also denote the set of units $U_x = \{u_{x_i} \mid i = 1 \cdots N_x\}$, and similary $U_y, U_z$ for set of all $u_{y_j}, u_{z_k}$ respectively. Each unit and the environment possess the unique varibles as listed in Table 1.

**Algorithm 1** Standard MLP

Initialize $w_{ij}, w'_{jk}$ for all $(i, j), (j, k)$
**while** criteria is not satisfied **do**
 **for** $d = 1 \cdots D$ **do**
 **for** $k = 1 \cdots N_z$ **do**
 $t_k \leftarrow t^{(d)}_{k_{data}}$
 **end for**
 **for** $i = 1 \cdots N_x$ **do**
 $x_i \leftarrow x^{(d)}_{i_{data}}$
 **end for**
 **for** $j = 1 \cdots N_y$ **do**
 $y_j \leftarrow \sigma(w_{0j} + \sum_{i=1}^{N_x} w_{ij} x_i)$
 **end for**
 **for** $k = 1 \cdots N_z$ **do**
 $z_k \leftarrow \sigma(w'_{0k} + \sum_{j=1}^{N_y} w_{jk} y_j)$
 $\delta_k \leftarrow \frac{2}{N_z}(z_k - t_k) z_k (1 - z_k)$
 **for** $j = 1 \cdots N_y$ **do**
 $w'_{jk} \leftarrow w'_{jk} - \eta \delta_k y_j$
 **end for**
 **end for**
 **for** $j = 1 \cdots N_y$ **do**
 $\delta_j \leftarrow y_j(1 - y_j) \sum_{k=1}^{N_z} w'_{jk} \delta_k$
 **for** $i = 1 \cdots N_x$ **do**
 $w_{ij} \leftarrow w_{ij} - \eta \delta_j x_i$
 **end for**
 **end for**
 $L^{(d)} \leftarrow \frac{1}{N_z} \sum_{k=1}^{N_z} (t_k - z_k)^2$
 **end for**
 $L \leftarrow \sum_{d=1}^{D} L^{(d)}$
**end while**

**Algorithm 2** General form of our multiagent algorithm (the environment)

Initialize $u_{x_i}, u_{y_j}, u_{z_k}$ for all $i, j, k$
**while** criteria is not satisfied **do**
 **for** all $d \in \{1 \cdots D\}$ **do**
 **for** all $u_{z_k} \in U_z$ **do**
 Input $t^{(d)}_{k_{data}}$ to $u_{z_k}$
 **end for**
 **for** all $u_{x_i} \in U_x$ **do**
 Input $x^{(d)}_{i_{data}}$ to $u_x$
 **end for**
 $L^{(d)} \leftarrow \sum_k (t_k - z_k)^2$
 **end for**
 $L \leftarrow \sum_d L^{(d)}$
**end while**

**Algorithm 3** Multiagent MLP (the environment)

Initialize $u_{x_i}, u_{y_j}, u_{z_k}$ for all $i, j, k$
**while** criteria is not satisfied **do**
 **for** $d = 1 \cdots D$ **do**
 **for** $k = 1 \cdots N_z$ **do**
 Input $t^{(d)}_{k_{data}}$ to $u_z$
 **end for**
 **for** $i = 1 \cdots N_x$ **do**
 Input $x^{(d)}_{i_{data}}$ to $u_x$
 **end for**
 $L^{(d)} \leftarrow \frac{1}{N_z} \sum_{k=1}^{N_z} (t_k - z_k)^2$
 **end for**
 $L \leftarrow \sum_{d=1}^{D} L^{(d)}$
**end while**

Table 1: Varibles each unit (and the environment) possess in multiagent MLP

| Unit | Variable |
| --- | --- |
| $u_{x_i}(i = 1 \cdots N_x)$ | $x_i$ |
| $u_{y_j}(j = 1 \cdots N_y)$ | $y_j, \delta_j, w_{ij}(i = 1 \cdots N_x), \eta$ |
| $u_{z_k}(k = 1 \cdots N_z)$ | $z_k, t_k, \delta_k, w'_{jk}(j = 1 \cdots N_y), \eta$ |
| The environment | $L, L^{(d)}$ |

The unit $u_{x_i}$ corresponds to $x_i$ and receive $x_{i_{data}}$ from the environment. The unit $u_{z_k}$ corresponds to both $z_k, t_k$ and is input $t_{k_{data}}$. Each $u_{x_i}$ receive input data, assign the data into the variable owned by itself, then send that value as message to $u_{y_j}$(algorithm 6).

$u_{y_j}$ corresponds to $y_j$ and stores $w_{ij}(i = 0 \cdots N_x)$ to calculate $y_j$. The unit must wait until receive message from all $u_{x_i}(i = 1 \cdots M)$. Then the unit can calculate $y_j$ (algorithm 4). The calculated value of $y_j$ is sent again to $u_{z_k}$, so similary $z_k$ can also become able to be calculated.

We also need to introduce a state variable for SDAE. We will discuss it at the end of this section.

Finally, the environment inputs data in the following order: $u_{z_1} \rightarrow \cdots \rightarrow u_{z_{N_z}} \rightarrow u_{x_1} \rightarrow \cdots \rightarrow u_{x_{N_x}}$ (Algorithm 3). These inputs stimulate units and the units invoke their message handling algorithm individually. The information required to calculate the cost function $L^{(d)}$ is eventually acumulated in the units $u_{y_k}$. The environment would read out these and calculate the objective $L^{(d)}$.

**Algorithm 4** The handler of $u_{y_j}$ when receiving a message

> **if** The sender is $u_{x_i}$ **then**
> > **if** Received messages from all $u_{x_i}(i = 1 \cdots N_x)$ **then**
> > > $y_j \leftarrow \sigma(w_{0j} + \sum_{i=1}^{N_x} w_{ij}x_i)$
> > > **for** $k = 1 \cdots N_z$ **do**
> > > > Send $u_{z_k}$ the value of $y_j$
> > >
> > > **end for**
> >
> > **end if**
> >
> **else if** The sender is $u_{z_k}$ **then**
> > **if** Received messages from all $u_{z_k}(z = 1 \cdots N_z)$ **then**
> > > $\delta_j \leftarrow y_j(1 - y_j) \sum_{k=1}^{N_z} w'_{jk}\delta_k$
> > > **for** $i = 1 \cdots N_x$ **do**
> > > > $w_{ij} \leftarrow w_{ij} - \eta\delta_j x_i$
> > >
> > > **end for**
> >
> > **end if**
> >
> **end if**

**Algorithm 5** The handler of $u_{z_k}$ when receiving data from the environment

> $t_k \leftarrow t_{k_{data}}^d$

**Algorithm 6** The handler of $u_{x_i}$ when receiving data from the environment

> $x_i \leftarrow x_{i_{data}}^d$
> **for** $j = 1 \cdots N_y$ **do**
> > Send $u_{y_j}$ the value of $x_i$ via message
>
> **end for**

**Algorithm 7** The handler of $u_{z_k}$ when receiving a message

> **if** Received messages from all $u_{y_i}(j = 1 \cdots N_y)$ and the environment **then**
> > $z_k \leftarrow \sigma(w'_{0k} + \sum_{j=1}^{N_y} w'_{jk}y_j$
> > $\delta_k \leftarrow \frac{2}{N_z}(z_k - t_k)z_k(1 - z_k)$
> > **for** $j = 1 \cdots N_y$ **do**
> > > $w'_{jk} \leftarrow w'_{jk} - \eta\delta_k y_j$
> >
> > **end for**
> > **for** $j = 1 \cdots N_y$ **do**
> > > Send $u_{y_j}$ the value of $w'_{jk}\delta_{z_k}$
> >
> > **end for**
>
> **end if**

Comparing both algorithms, we can verify that the all update statement and order for all variables are strictly matched between the proposed multiagent SDAE and the standard version.

### 3.3.2 EQUIVALENCE WITH AUTOENCODER

Now we advance to the autoencoder algorithm which is a special case of MLP. In autoencoders, The input and output dimension is the same ($N_x = N_z$). The cost function and difference at output units are changed to $L^{(d)} \leftarrow \frac{1}{N_z} \sum_{k=1}^{N_z} (x_k - z_k)^2$ and $\delta_k \leftarrow \frac{2}{N_z}(z_k - x_k)z_k(1 - z_k)$ respectively. In the corresponding multiagent version, the environment don't need to send $t_{i_{data}}$ to $u_{z_i}$. Instead $u_{x_i}$ send the input value $x_i$ to $u_{z_i}$ right before messaging to $u_{y_1}$.

$u_{z_i}$ don't take any action in response to receiving message from $u_{x_i}$, just as same as from the environment. With the modifications above, the two versions become equivalent.

### 3.3.3 EQUIVALENCE WITH DENOISING AUTOENCODER

Denoising Autoencoder (hereafter "DAE") is an extension of autoencoder and used as a building block of SDAE. In DAE, when calculating $y$ of the hidden layer, we don't directly use raw value of $x$. Instead, we add noise to $x$ to make $\tilde{x}$, then $y$ uses $\tilde{x}$. Note that the reconstruction layer $z$ still use the original $x$ value.

To reinterpret this DAE by our multiagent system, we can assign $\tilde{x}_i$ to $u_{x_i}$ along with $x_i$. When the data is input, we can calculate $\tilde{x}$ right after the assignment to $x_i$. Each $x_i(i = 1 \cdots N_x)$ send corresponding $y_j(j = 1 \cdots N_y)$ the noised value $\tilde{x}_i$ instead of $x_i$. Then the two algorithm become equivalent again.

### 3.3.4 Equivalence with standard SDAE

We finally reimplement the whole SDAE algorithm by our multiagent system and show its equivalence with the standard version again. The learning process of SDAE can be divided into pretraining and fine-tuning and we can explain the two phase separately.

**pretraining phase** In the pretraining phase of SDAE, let $x_i, y_i, z_i$ be the input, hidden, the associated reconstruction layer value of DA, respectively. To stack a learning DA, $y_j$ is considered as a new input units from the view of the new DA. Let $z'_j$ be the reconstruction unit's value of $y_j$ itself, and $y'_k$ be the new stacked hidden layer units. We make a full connection between each layers. Thus the equations relating to these variables change to $y'_k \leftarrow \sigma(w_0 + \sum_{j=1}^{N_y} w_{jk} y_j)$ and $z'_j \leftarrow \sigma(w_0 + \sum_{k=1}^{N_{y'}} w_{kj} y'_k)$. The new objective function after the stacking is $L_j = \frac{1}{N_y} \sum_{j=1}^{N_y} (z'_j - y_j)^2$. Note that by stacking a new DAE layer, we are making a new task of objective function from the previous objective.

Now we describe the change needed on our multiagent system. We need to introduce the notion of state for the units associated with the hidden layers in the standard version. When the objective function changed, the network structure and the state of the units must be changed in response too.

The hidden units can take the the two states, "state A (On learning)" and "state B (Learned, stable)". The units are initiated to "state A" and the behavior in this state doesn't change from the previous sections. But the units in "state B" doesn't send messages to the units for reconstruction $z_i$ and instead send the same messages to the "state A" units in the stacked layer.

Additionary, we need to append a reconstruction unit for each hidden unit changed to "state B". These pairs of the "state B" unit and appended reconstruction unit act as same as the pairs of input and reconstruction units we stated in DAE section except that they send message not after the data is input, but after the feature $y_j$ is calculated.

Once the pretraining for one block of DAE ends, all the hidden units in that DAE change to "state B". Then they form a new DAE starting from "state B" hidden units through the new stacked "state A" units then the associated reconstruction units appended. This DAE can be optimized in the same way of the single multiagent DAE we have described with only one exception that the data is not sent from the environment, but instaed the feature value $y_j$ in the previous DAE is calculated in the same order ($j = 1 \cdots N_y$)

**fine-tuning phase** In order to use SDAE for supervised learning task, we pretrain some predefined number of hidden layers by layerwise pretraining, and at the end stack the layer for supervised purpose (typically a softmax layer for classification). This is another form of the change of the objective function and the multiagent system can deal with it too by changing network and unit's state.

Specifically, when the all pretraining phase ends, connect the output units for supervised learning onto the topmost hidden units, and all feature unit changes its state to the newly introduced "state C (On fine-tuning)". The "state C" is almost the same as "state A", but the only differences are the two:

- They don't send message to reconstruction units.
- They always send backpropagation message to the previous units.

This backpropagation message must be sent in ascending order as well as the previous message passing.

Here we reinterpreted the standard pretraining and fine-tuning algorithm with our proposed multiagent system. These two cover the all learning phase of SDAE, therefore we have successfully reinterpreted the whole SDAE and showed its equivalence of the standard SDAE.

### 3.4 Sparse connect SDAE

Now we obtain new multiagent SDAE, and theoritically showed its equivalence to standard SDAE. Next we want to extend this model and experiment its behavior. One of the minimal and simplest

modification to this model is to truncate the edges randomly. We call this model "Sparse connect SDAE" or "SCSDAE" for the latter section.

Since our goal here is to check the basic behavior when we truncate edges, we take one of the simplest rule that we basically connect all edges between nodes (if the distance meets criteria) but drop them for certain threshold probability. Once the connection is established, we fix them and will not modify online. For example, connection rate 1.0 (100%) means it is the same network as SDAE and 0.3 means 30% of edges in SDAE remain intact and the others are dropped.

We can expect the time spent in learning is proportional to the connection rate. This is an obvious strong point against the standard matrix-vector based network. In matrix-vector based networks, we must fix the size of weight matrices and keep them at their original size even if most weights are learned to be soft zero. In our multiagent model however, we only need to calculate for existing edges. We will see whether this expectation holds in the next experiments section.

## 4 EXPERIMENTS

We implemented the proposed multiagent network and measure its performance. We verified empirical equivalence between the multiagent SDAE and standard SDAE. We also measure performance of Sparse connect SDAE, which we described in 3.4. We change its connection rate and see how the performance and learning time changes.

### 4.1 DATASET

We use two datasets for experiment, one is an artificial XOR function toy dataset made by us for this experiment, and the other is the MNIST handwritten digit dataset. Since the result and conclusion is almost same, we only report for the MNIST dataset. The details of the datasets are described in Appendix A.

### 4.2 MODEL SETTINGS

We use the sigmoid function as activation functions of all units. We use the unit-wise MSRE as mentioned in the section 3 for both reconstruction errors in pretraining phase and classification in finetune phase. In pretraining, we simply take the mean through the all reconstruction units to check if the training works. For finetuning, we again take the sum of MSRE as cost function, and as classification error, we take the label associated with the unit that has the most activated value and discretely compare that value to measure the error rate. The SDAE algorithm includes random noise addition and is not fully deterministic. So we run each experiment settings for 3 times and take the average through the runs for all metrics we show. We set learning rate for pretraining and finetuning to 0.001 and 0.1, respectively. Corruption rate of SDAE is fixed to 0.3.

### 4.3 COMPARISON BETWEEN MULTIAGENT AND NORMAL SDAE

We compared the test error rate through epochs between multiagnt and normal SDAE. Figure 2 show the mean classification error rate through 3 simulations with the MNIST dataset. We can see the two graphs form almost the same shape, which suggests the multiagent implementation of SDAE is empirically equivalent to normal SDAE. This is a verification of the theoritical proof we showed in section 3.

In order to check this equivalence more precisely, we evaluate the difference of maximum weight update between multiagent and normal SDAE (Table 2). We gave an input data to both networks and measure the largest change of weight change update for each edge group shown in Table 2. To make the problem simple and clear, for this experiment we used only one hidden layer and no pretraining, so the model is similiar to a naive MLP. From the Table 2, we can see that the decimal order of largest difference is 3 digits small as that of information error with 32bit floating point numbers. We suppose this difference is small enough to be well ignored.

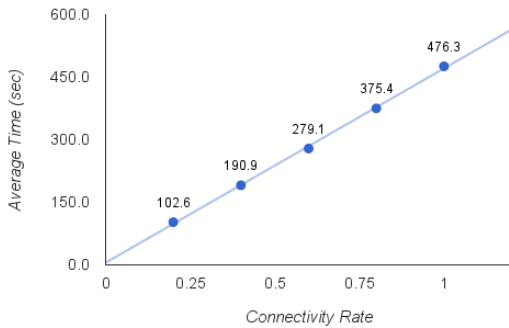

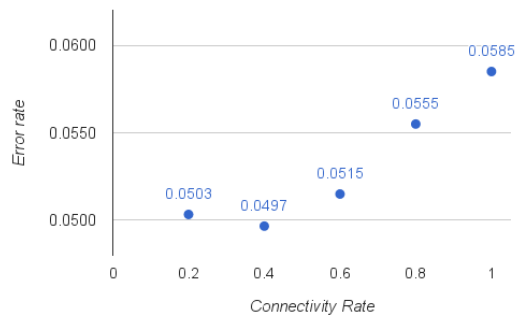

Figure 3: Mean calculation time for connectivity rate averaged over 3 experiments for each connectivity. The horizontal axis indicates connectivity rate of each model and the vertical shows time spent in training.

Figure 4: Mean error rate for connectivity rate averaged over 3 experiments for each connectivity. The horizontal axis indicates connectivity rate of each model and the vertical for error rate in test dataset.

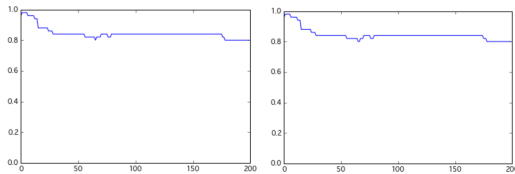

Figure 2: Mean classification error rate (left: standard SDAE, right: multiagent SDAE). The vertical coordinate indicates their error rate and the horizontal for epoch count. Both SDAE result in almost the same shape.

Table 2: Difference of the largest weight change for one input between multiagnet and normal SDAE

| Group of edges | Largest difference |
|---|---|
| input to hidden | $1.08 \times 10^{-19}$ |
| hidden bias | 0.0 |
| hidden to output | $8.7 \times 10^{-19}$ |
| output bias | 0.0 |

### 4.4 SPARSE CONNENCT SDA

We gradually changed the connectivity rate of our proposed SCSDAE and compare its performance. In this experiment, we measure the performance by the two aspect, the error rate and the calculation time, because we expect that the less number of edge mean the less calculation in time in our model as we described in chapter 3.

Figure 3 shows the meausred calculation time and Figure 4 for the error rate. In both figures the horizontal coordinate indicates the number of iterated epochs and the vertical is for time and error rate, respectively. From Figure 3, we can verify that the calculation time decreases linearly as we expected.

Figure 4 shows the error rate on test set after the given number of epochs passed. Contrary to our intuition, it can be said that in some cases despite of the decrease in connection probability, error rate do not deteriorate. The reason is not obvious, but we suppose that the more connection established, the more it became difficult and time requiring to the model to converge enough to show the potential of the model class. It can be also possible that the forced sparsity gave the model unexpected advantage to obtain sparse coding efficiently.

## 5 CONCLUSION

We proposed a fundamental framework to reinterpret deep networks as multiagent systems. Specifically, we reimplemented a model equivalent to Stacked Denoising Autoencoder(SDAE) but the inside implementation is given with the multiagent system. We verified this equivalence from both theoritical and empirical aspects. We also tested the behaviour when we actually let the model drop

some edges permanently and demostrated the reduction of computation time as the connection rate decreases.

Our contribution is to propose new multiagent based neural network system that free existing deep network from restriction of layer scheme. Our system involves the standard SDAE as its subset and has large potential of extension. Our model could be extended to more biologically plausible variation. Our experiment with the proposed Sparse Connect SDAE demostrate the advantage of non-matrix calculation and permanent drop of edges.

The next step is to sophisticate the model so that the agents are more strictly independent from the environment. Then we will be able to use this system as a framework for extending more flexible deep network. For example, we can make arbitrary connection of units. We can also consider mixing of units with different activation function, dropout strategy, learning rate.

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

APPENDIX

A. DATASETS

XOR function toy dataset consists of random ordering of four points representing the four region of XOR function on 2D coordinate, which are set of tuple $(x_1, x_2, y)$ = $[(1, 1, 1), (1, -1, -1), (-1, 1, -1), (-1, -1, -1)]$. The value '1' represents boolean value 'true' and '-1' for 'false'. $x_1$ and $x_2$ mean boolean value input to the XOR function and $y$ is the output.

MNIST handwritten digit dataset is a famous image recognision task. The input is a 28x28 image patch of digit classified to 10 classes(0-9). It contains 60000 train images and 10000 test images. Since our goal is not to pursue the state of the art performance of this dataset, but verify the common property of our proposed model with more realistic problem rather than the XOR toy sample, so we did no preprocessing on the dataset. We directly input image pixel value as a 784 row vector and didn't use the prior that the image is 2D data with spatiality which is importatnt in convolutional network. This setting is called "permutation invariant" version of the MNIST task.

B. EXPERIMENTS DETAILS

For the XOR task, we set the number of hidden units to [3, 4, 5]. For MNIST, we used [100, 100, 100] network for all experiments. The learning rate was fixed to 0.001 at pretraining time and 0.1 at finetuning. We trained 30 epochs of pretrainings for each MLP layer and 50 epochs of finetuning. We used a single core of Intel(R) E5-2630 @ 2.40GHz to measure the performance invariant to paralization method. The effect of paralization is a question for future research.

