# Peer review of "Multiagent System for Layer Free Network"

_ICLR 2017 — rejected_

[Official Review · AnonReviewer1 · rating 2 · confidence 5 · 19 Dec 2016]
**No Title**

Unfortunately, the paper is not clear enough for me to understand what is being proposed. At a high-level the authors seem to propose a generalization of the standard layered neural architecture (of which MLPs are a special case), based on arbitrary nodes which communicate via messages. The paper then goes on to show that their layer-free architecture can perform the same computation as a standard MLP. This logic appears circular. The low level details of the method are also confusing: while the authors seem to be wanting to move away from layers based on matrix-vector products, Algorithm 4 nevertheless resorts to matrix-vector products for the forward and backwards pass. Although the implementation relies on asynchronously communicating nodes, the “locking” nature of the computation makes the two entirely equivalent.

[Official Review · AnonReviewer3 · rating 1 · confidence 5 · 20 Dec 2016]
**Lack of novelty and of knowledge of the relevant literature and development history**

The paper reframes feed forward neural networks as a multi-agent system.

It seems to start from the wrong premise that multi-layer neural networks were created expressed as full matrix multiplications. This ignores the decades-long history of development of artificial neural networks, inspired by biological neurons, which thus started from units with arbitrarily sparse connectivity envisioned as computing in parallel. The matrix formulation is primarily a notational convenience; note also that when working with sparse matrix operations (or convolutions) zeros are neither stored not multiplied by.

Besides the change in terminology, essentially renaming neurons agents, I find the paper brings nothing new and interesting to the table.

Pulling in useful insights from a different communitiy such as multi-agent systems would be most welcome. But for this to be compelling, it would have to be largely unheard-of elements in neural net research, with clear supporting empirical evidence that they significantly improve accuracy or efficiency. This is not achieved in the present paper.

[Official Review · AnonReviewer2 · rating 3 · confidence 3 · 25 Dec 2016]
**No Title**

The multiagent system is proposed as a generalization of neural network. The proposed system can be used with less restrictive network structures more efficiently by computing only those necessary computations in the graph. Unfortunately, I don't find the proposed system different from the framework of artificial neural network. Although for today's neural network structures are designed to have a lot of matrix-matrix multiplications, but it is not limited to have such architecture. In other words, the proposed multiagent system can be framed in the artificial neural network with more complicated layer/connectivity structures while considering each neuron as layer. The computation efficiency is argued among different sparsely connected denoising autoencoder in multiagent system framework only but the baseline comparison should be against the fully-connected neural network that employs matrix-matrix multiplication.

[Final Decision · Program Chairs · 06 Feb 2017]
**ICLR committee final decision**

The reviewers were consistent in their review that they thought this was a strong rejection.
 Two of the reviewers expressed strong confidence in their reviews.
 The main arguments made by the reviewers against acceptance were:
 Lack of novelty (R2, R3)
 Lack of knowledge of literature and development history; particularly with respect to biological inspiration of ANNs (R3)
 Inappropriate baseline comparison (R2)
 Not clear (R1)
 
 The authors did not provide a response to the official reviews. Therefore I have decided to follow the consensus towards rejection.